# Accelerated, Dose escalated, Sequential Chemoradiotherapy in Non-small-cell lung cancer (ADSCaN): a protocol for a randomised phase II study

Matthew Q F Hatton,[1] Claire Anne Lawless,[2] Corinne Faivre-Finn,[3] David Landau,[4] Jason F Lester,[5] John Fenwick,[6] Rita Simões,[7] Elaine McCartney,[2] Kathleen Anne Boyd,[8] Tom Haswell,[9] Ann Shaw,[2] James Paul[2]

For numbered affiliations see end of article.

**Correspondence to**
Dr Matthew Q F Hatton;
matthew.hatton@sth.nhs.uk

## ABSTRACT

**Introduction** Lung cancer is the most common cause of cancer mortality in the UK, and non-small-cell lung cancer (NSCLC) accounts for approximately 85% of all lung cancers. Most patients present with inoperable disease; therefore, radiotherapy plays a major role in treatment. However, the majority of patients are not suitable for the gold standard treatment (concurrent chemoradiotherapy) due to performance status and comorbidities. Novel strategies integrating radiotherapy advances and radiobiological knowledge need to be evaluated in patients treated with sequential chemoradiotherapy. Four separate dose escalation accelerated radiotherapy schedules have been completed in UK (CHART-ED, IDEAL-CRT, I-START and Isotoxic IMRT). This study will compare these schedules with a UK standard sequential chemoradiotherapy schedule of 55 Gy in 20 fractions over 4 weeks. As it would be impossible to test all schedules in a phase III study, the aim is to use a combined randomised phase II screening/'pick the winner' approach to identify the best schedule to take into a randomised phase III study against conventionally fractionated radiotherapy.

**Methods and analysis** Suitable patients will have histologically/cytologically confirmed, stage III NSCLC and are able to undergo chemoradiotherapy treatment. The study will recruit 360 patients; 120 on the standard arm and 60 on each experimental arm. Patients will complete 2–4 cycles of platinum-based chemotherapy before being randomised to one of the radiotherapy schedules. The primary endpoint is progression-free survival, with overall survival, time to local–regional failure, toxicity and cost-effectiveness as secondary objectives.

**Ethics and dissemination** The study has received ethical approval (research ethics committee (REC) reference: 16/WS/0165) from the West of Scotland REC 1. The trial is conducted in accordance with the Declaration of Helsinki and Good Clinical Practice. Trial results will be published in a peer-reviewed journal and presented internationally.

**Trial registration number** ISRCTN47674500.

## Strengths and limitations of this study

► Radiotherapy delivered using advanced techniques, including four-dimensional CT and image-guided radiotherapy. Intensity-modulated radiotherapy is mandated in one arm and recommended in all others.
► Robust quality assurance programme.
► Innovative multiarm 'pick the winner' design.
► Not all experimental arms are available at all sites, so although arms are randomised against control, they are not randomised against each other.
► Final analysis relies on adjusting for between-centre differences in key patient characteristics in order to 'pool' the control arm and achieve the target power.

## INTRODUCTION
### Background

Lung cancer is the most common cause of cancer mortality in the UK. In 2014, there were 46 400 new cases and 35 900 deaths and non-small-cell lung cancer (NSCLC) accounts for approximately 85% of these. Most cases are inoperable at presentation; however, many patients with stage I–III disease can be treated radically using radiotherapy (RT), although survival rates are disappointing. Successful local control has been found to correlate with improved survival.[1] Strategies to achieve improved local control rely on intensification of local antitumour effect through radiation sensitisation, acceleration of RT schedule or dose escalation. As most patients are not suitable for gold standard treatment of concurrent chemoradiotherapy, novel strategies integrating RT technological advances and radiobiological knowledge need evaluating in patients treated with the alternative option of sequential chemoradiotherapy. The UK strategy has been to develop separate escalation protocols for hyperfractionated

and once-daily schedules (CHART-ED,[2] IDEAL-CRT,[3] I-START[4] and Isotoxic IMRT[5]). These studies have completed recruitment and the results of two have been published.[1 2]

### Dose acceleration/intensification

In the UK, Continuous Hyper-fractionated Accelerated Radiation Therapy (CHART) trial[6] intensified local treatment by accelerating the RT course. When compared with conventional fractionation, there was a 9% absolute improvement in 2-year survival (29% vs 20%; p=0.004) which provides evidence that intensification of local treatment only results in increased overall survival (OS). This result is supported by meta-analysis[7] reporting an absolute benefit for hyperfractionated or accelerated RT over conventional fractionations of around 3% at 5 years.

Further evidence supporting intensification of local treatment comes from trials showing a survival benefit of concurrent chemoradiotherapy over sequential.[5] This was achieved by chemotherapy increasing local tumour cell kill and improving local control while maintaining the systemic effect of chemotherapy.

### RT dose escalation

Increasing the RT dose has been associated with seemingly improved local control and OS in early-phase studies. However, Radiation Therapy Oncology Group 0617[8] comparing 60 Gy in 30 fractions of radiotherapy to 74 Gy in 37 fractions of radiotherapy once-daily showed that local control and OS were poorer on the higher dose. Reasons for this are being debated, but it is clear that increasing RT by 2 Gy per fraction per day with an increase in overall treatment time will not improve outcomes.

### Isotoxic RT

The MAASTRO group pioneered the concept of 'isotoxic RT' allowing for individualised dose escalation using hyperfractionated accelerated RT based on predefined and mean lung dose in stage I–III patients. This showed with three-dimensional conformal RT delivered two times a day over 4 weeks that increasing the radiation dose to prespecified normal tissue dose constraints could lead to increased tumour control probability (TCP) with the same normal tissue complication probability.[9] Survival of these stage III patients, all of whom were treated sequentially, was comparable to results expected from concurrent chemoradiotherapy with acceptable acute and late toxicity. Since this study reported, intensity-modulated radiotherapy (IMRT) techniques have become routine practice improving targeting of radiation doses while decreasing doses to lungs and other normal tissue. This gives the potential to use the isotoxic approach to escalate the dose delivered to the target increasing TCP with the same normal tissue complication probability.[10]

### Dose acceleration and escalation

Techniques that avoid prolongation of overall treatment time are attractive as they reduce the impact of accelerated tumour clonogen proliferation, which becomes clinically relevant for NSCLC approximately 3–4 weeks after RT initiation. A meta-analysis[7] of altered fractionations (10 trials, 2000 patients with NSCLC) showed modified fractionation that improved OS as compared with conventional schedules (HR=0.88, 95% CI 0.80 to 0.97; p=0.009), resulting in an absolute benefit of 2.5% at 5 years. CHART (54 Gy in 36 fractions of radiotherapy over 12 days) remains one of the more commonly used schedules for radical treatment of NSCLC in the UK along with daily fractionated schedules delivering 55 Gy in 20 fractions of radiotherapy over 4 weeks or 60–66 Gy in 30–33 fractions of radiotherapy over 6–6½ weeks.[11] In current UK practice, both CHART and the 4-week schedule are deliverable, with 99% of patients completing treatment, with survival outcome matching that seen in the original CHART study and very low levels of grade III/IV toxicity.[12] However, survival rates for all three regimens are low and could be improved by dose intensification.

Radiobiological modelling suggests that dose escalation to improve the TCP is likely to be more effective if the overall treatment time is fixed rather than fixing the dose per fraction, and dose escalation typically adds 1%–2% local control for each 1% increase in dose.[13] Therefore, Cancer Research UK (CRUK) funded four parallel dose-escalation studies to intensify accelerated schedules used in the UK. The first, CHART-ED (ISRCTN 45918260)[1] reported no dose-limiting toxicities with dose escalation (using additional twice-daily fractions of 1.8 Gy to 64.8 Gy in 17 days). This dose increase has the potential to increase local tumour control by 30%. The IDEAL-CRT trial (ISRCTN 12155469) evaluated dose escalation up to 73 Gy in 30 fractions of radiotherapy over 6-week schedule with dose escalation calculated on an individual patient basis according to either lung or oesophageal radiation dose. The study has established 68 Gy to 1 cc oesophagus as the maximum tolerated dose (MTD) to be taken forward into future trials.[2] Both CHART-ED and IDEAL-CRT report encouraging 2-year survival figures: 52% and 65%, respectively.

Following the confirmation of low toxicity with the original IDEAL-CRT trial, a further accelerated schedule was investigated where overall treatment time was shortened from 6 to 5 weeks by treating with RT twice a day on 1 day/week (not on chemotherapy day). The MTD to 1 cc oesophagus was reduced to 65 Gy and the maximum prescription dose allowed was reduced to 71 Gy in 30 fractions of radiotherapy (over 5 weeks). This portion of the study recruited 36 patients with a minimum 6-month follow-up and no excess toxicity. This schedule will be taken forward and given sequentially in ADSCaN (Accelerated, Dose escalated, Sequential Chemoradiotherapy in Non-small-cell lung cancer).

The two remaining studies, I-START (ISRCTN 74841904),[3] which individualised the RT dose similarly to IDEAL-CRT using a schedule following sequential chemotherapy in 20 fractions of radiotherapy, and Isotoxic IMRT (NCT01836692),[4] where patients were

treated twice daily over 4 weeks to a maximum of 79.2 Gy using prespecified normal tissue doses have completed recruitment and neither report excess toxicity (personal communication).

The gold standard treatment for stage III NSCLC is concurrent chemoradiotherapy.[4] However, in practice guidelines can only loosely define eligibility for the concurrent approach[14] and for many patients the concerns around fitness and the toxicity of this approach lead to chemoradiotherapy being given sequentially.[15] A recent analysis of the National Cancer Database[16] suggests there is a basis for these concerns and that there are categories of patients for whom the higher toxicity levels associated with the concurrent approach could prove detrimental. At this point of time, those categories are ill defined and novel strategies that integrate RT technological advances and radiobiological knowledge need evaluating in patients treated with sequential chemoradiotherapy. This study aims to provide a framework for further development by identifying which is the best accelerated schedule to take forward into a randomised phase III study against conventionally fractionated RT.

## METHODS AND ANALYSIS

ADSCaN is a non-blinded multicentre phase II study sponsored by NHS Greater Glasgow and Clyde Health Board and coordinated by the CRUK Clinical Trials Unit (CRUK CTU) based at the University of Glasgow. It is registered on the ISRCTN clinical trials database (ISRCTN47674500) and funded by CRUK's Clinical Trials Awards and Advisory Committee (Funder Reference: C9759/A16604). RT quality assurance (QA) is supported by the National Radiotherapy Trial Quality Assurance (RTTQA) group. The study is included in the National Institute for Health Research (NIHR) Clinical Research Network portfolio (ID 35624) and is being conducted in accordance with the Declaration of Helsinki and Good Clinical Practice.

We hypothesise that sequential chemoradiotherapy using accelerated, dose-escalated RT will improve progression-free survival (PFS), OS, time to local–regional failure, toxicity and cost-effectiveness compared with conventionally fractionated sequential chemoradiotherapy treatment. ADSCaN will take four dose-escalated accelerated sequential chemoradiotherapy schedules into a randomised phase II comparison with a UK standard sequential chemoradiotherapy using state-of-the art RT.

### Patient and public involvement

Patient and carer representatives have been involved from the early stages of development of the ADSCaN study. These representatives had personal experience of sequential chemoradiotherapy treatment and ensured the study reflected their priorities in developing the most appropriate chemoradiotherapy regimens for this cohort of patients. Their engagement has been maintained through the National Cancer Research Institute

Lung Clinical Studies Group and the West of Scotland Cancer Research Network Consumer Research Panel. In particular, we thank Tom Haswell who has become our consumer representative on the Trial Management Group (TMG) to help oversee recruitment and conduct of the study.

Detailed results of the study will be given the study participants/their relatives on request and disseminated on the publically accessible websites of our funders, CRUK, and other supportive charities (eg, Roy Castle Lung Cancer Foundation).

### Setting

ADSCaN opened on 22 August 2017 and will recruit 360 patients with a histological/cytological proven NSCLC across 40 UK RT centres, a list of which can be found on the NIHR Clinical Research Network Portfolio. Patients will be stage III, performance status (PS) 0–1 and unsuitable for concurrent chemoradiotherapy. Individual centres will randomise patients between standard RT (55 Gy in 20 fractions of radiotherapy) and one or more experimental arms. All sites will offer the standard arm, but may choose which experimental arms they wish to use (figure 1: flow diagram).

### Participant screening and selection

Patients deemed inoperable by the lung MDT and suitable for sequential chemoradiotherapy are invited to participate and provided with a patient information sheet (PIS, online supplementary appendix 1). Mandatory prerandomisation investigations include: lung function tests, contrast CT scan of thorax/upper abdomen within 4 weeks prior to randomisation and fluorodeoxyglucose PET with CT (or MRI) brain scan if patients have not these investigations prior to starting induction chemotherapy.

### Inclusion criteria

► Histologically/cytologically confirmed stage III NSCLC.
► ECOG PS 0–2 (PS 2 only included if the local investigator deems general condition is explained by disease or the primary chemotherapy treatment).
► Inoperable disease.
► Inoperable disease, unsuitable for concurrent chemoradiation, in the opinion of the treating oncologist.
► Patients with complete response, partial response or stable disease on CT assessment after 2 cycles of platinum-based chemotherapy.
► Aged ≥16 years.
► Adequate pulmonary function test results: forced expiratory volume in 1 s and/or diffusing capacity of lung for carbon monoxide transfer coefficient ≥40% of predicted.

### Exclusion criteria

► Previous/current malignant disease likely to interfere with the protocol treatment or comparisons.
► Clinically significant interstitial lung disease.

## ADSCaN Trial Schema

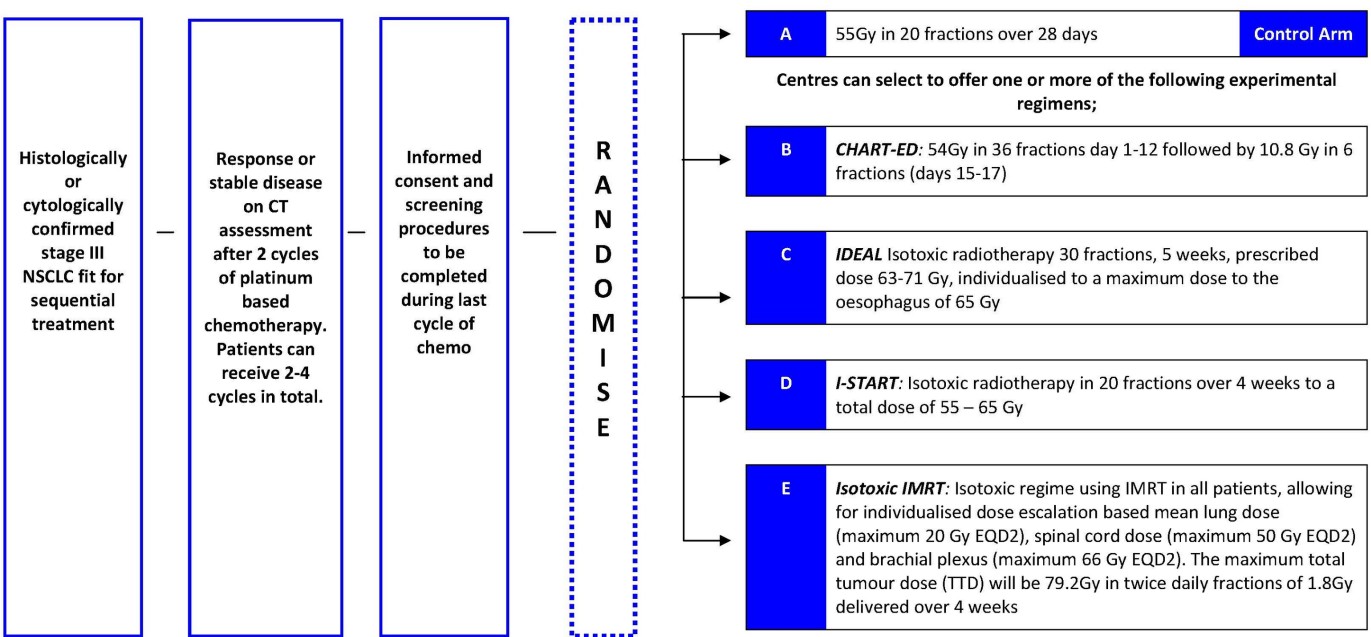

ADSCaN Trial Schema
Version 5.0, 17Jan2017

**Figure 1** ADSCaN trial schema. ADSCaN, Accelerated, Dose escalated, Sequential Chemoradiotherapy in Non-small-cell lung cancer; EQD2, equivalent dose for 2 Gray fractions; IMRT, intensity-modulated radiotherapy; NSCLC, non-small-cell lung cancer.

## Informed consent

Eligibility is confirmed by a clinician and patients are given ≥24 hours to consider the PIS and ask questions prior to written informed consent being taken.

## Randomisation

Randomisation occurs after completion of chemotherapy. Once a patient is deemed eligible for the study and has consented to participate, the CRUK CTU, Glasgow, is contacted to confirm eligibility, and subsequently the patient is randomised to one of the available ADSCaN arms offered on-site and issued with a participant identification number. Patient enrolment details are confirmed by email.

Minimisation incorporating a random factor is used to allocate patients between treatment arms A:B:C:D:E so that an overall study ratio of 2:1:1:1:1 is achieved. The following stratification factors are used:

► Hospital.
► PS.
► Intention to use IMRT.
► Histological type.
► Stage.
► Epidermal growth factor receptor mutation status.
► Gross tumour volume (GTV).
► Haemoglobin.
► Serum albumin.
► Sex.

► Weight loss since diagnosis.

While on study, participants will not co-enrol in other clinical trials offering therapeutic intervention.

## Standard care

In patients with inoperable stage III NSCLC who are unsuitable for concurrent chemoradiotherapy, sequential chemoradiotherapy is the standard of care.

Patients who are eligible for this trial will receive 2–4 cycles of platinum doublet neoadjuvant chemotherapy. The recommended non-platinum agents are pemetrexed or vinorelbine for non-squamous carcinoma and gemcitabine or vinorelbine for squamous carcinoma. TMG/DMC (Data Monitoring Committee) are aware that the standard of care for the systemic treatment of stage III NSCLC may evolve over the course of this study and will consider the evolving evidence on newer approaches such as immunotherapy.

## RT intervention and planning

Four-dimensional CT with intravenous contrast should be used for all ADSCaN patients to account for tumour motion during breathing. The whole thorax (cricoid to L2) should be covered to allow dose volume histograms to be calculated for the lung, heart, spinal cord, brachial plexus, great vessels, proximal bronchial tree and the oesophagus.

**Table 1** Summary of trial arms

| Trial arm | Total dose (Gy) | Total fractions | Fractions per day | Total number of days |
|---|---|---|---|---|
| Arm A: Standard | 55.0 | 20 | 1 | 26–28 |
| Arm B: CHART-ED | 54.0 | 36 | 3 | Days 1–12 |
| | 10.8 | 6 | 2 | Days 15–17 |
| Arm C: IDEAL-CRT | 63–71 | 30 | 1 with 1 twice daily treatment per week | 33–35 |
| Arm D: I-START | 55–65 | 20 | 1 | 26–28 |
| Arm E: Isotoxic IMRT | 61.2–79.2 | 34–44 | 2 | 23–30 depending on dose escalation |

The input of a thoracic radiologist is recommended when defining the identifiable tumour and involved lymph nodes to give the GTV. The thoracic radiologist should work with a qualified radiation oncologist specialised in thoracic malignancies to produce a motion-adapted GTV.

The 4D-GTV is expanded by 5 mm to create the clinical target volume (CTV). The CTV will be expanded to include microscopic spread and may be edited to account for anatomical barriers but only if it is not thought to invade the structure.

CTV to planning target volume (PTV) expansion takes into account patient set-up uncertainties:

► If a centre is using per-fraction cone beam CT (CBCT) imaging and online correction for treatment verification, the CTV should be expanded by 5 mm isotropically to create the PTV.

► If a centre is not CBCT imaging per fraction, the CTV should be expanded by 7 mm axially and 9 mm cranio/caudally to create the PTV.

### Prescribed dose and fractionation
RT schedules should begin on a Monday where possible to reduce overall treatment time. Schedules are summarised in table 1, and dose constraints for organs at risk (OAR) are given in table 2.

All patients recruited at a centre must be treated with the same planning technique irrespective of randomisation arm. The use of IMRT is strongly recommended (and mandated for arm E) using five or more fields in an IMRT plan or an arc technique may be used.

### Follow-up
Patients are followed up monthly for 3 months, then 3-monthly to 2 years, 6-monthly to 3 years and then annually. A late toxicity assessment will be performed at each visit.

### Withdrawal
Patients can withdraw from the trial at any time without any effect on clinical care.

**Table 2** Dose constraints for all trial arms

| Volume | Arm A: Standard | Arm B: CHART-ED | Arm C: IDEAL-CRT | Arm D: I-START | Arm E: Isotoxic IMRT |
|---|---|---|---|---|---|
| CTV | V95%>99% | V95%>99% | V95%>99% | V95%>99% | Optimal: CTV D(total Vol −1 cc) ≥95% Mandatory: CTV$^{MeanDose}$±1% of prescribed dose |
| PTV | V95%>90% V90%>98% | V95%>90% V90%>98% | V95%>90% V90%>98% | V95%>90% V90%>98% | Optimal: PTV D(total vol −1 cc) ≥90% PTV D95%≥95% Mandatory: PTV D(total vol − 1 cc)≥85% PTV D95%≥90% PTV D1 cc≤107% |
| Spinal canal | D0.1 cc<47 Gy | D0.1 cc<46 Gy | D0.1 cc<48 Gy | D0.1 cc<47 Gy | D1.0 cc ≤ EQD2 50 Gy |
| Lung GTV | V20 Gy<35% | V20 Gy<35% | V20 Gy<35% | | Mean dose≤20 Gy (physical dose) |
| Brachial plexus | D0.1 cc<55 Gy | D0.1 cc<64.3 Gy | D0.1 cc<65 Gy D30%<60 Gy (ipsilateral branchial plexus) | D0.1 cc<55 Gy | D1.0 cc ≤ EQD2 66 Gy |
| Heart | D100%<36 Gy D67%<44 Gy D33%<57 Gy | D100%<48 Gy D67%<57 Gy | D100%<44 Gy D67%<52 Gy D33%<59 Gy | D100%<36 Gy D67%<44 Gy D33%<57Gy | D1.0 cc ≤ EQD2 76 Gy Mean dose≤46 Gy (physical dose) |
| Oesophagus | D0.1 cc<105% | D0.1 cc<105% | D1.0 cc<65 Gy | | |
| Lung contra | V5 Gy<60%* | | V5 Gy<60%* | V5 Gy<60%* | |
| Mediastinal envelope | | | | | D1.0 cc ≤ EQD2 76 Gy |

*Optimal constraint (ie, not mandatory limit).
CTV, clinical target volume; EQD2, equivalent dose for 2 Gray fractions; GTV, gross tumour volume; IMRT, intensity-modulated radiotherapy; PTV, planning target volume.

## Quality of life

Quality of life (QoL) as measured by EuroQol-5D (EQ-5D) will be completed at all study visits and weekly during treatment.

## Data collection

All information collected during the course of the study will be kept strictly confidential. Data will be entered on to an electronic case report form system with built-in edit checks to ensure high-quality data. Sites will be monitored centrally by checking incoming data for compliance with the protocol, data consistency, missing data and timing.

## Statistical considerations

The comparison of each experimental arm to standard treatment has 80% power to detect a 33% increase in median PFS (from 8 to 10.7 months) at the 20% one-sided level of statistical significance.[17] A 33% improvement in PFS was selected as it was felt realistic and that an improvement of this magnitude was required to ensure UK support for any subsequent phase III. There is no formal calculation of sample size for the 'pick-the-winner' element; if more than one experimental arm passes the screening threshold, a choice will be made using efficacy, toxicity, QoL and health economics (HE) evidence.

### Primary analysis

The primary statistical analysis will be based around a Cox model fitted to PFS incorporating terms for study arm and other prognostic factors that reflect case mix. Model fit and the proportional hazards assumption will be checked by plotting Schoenfeld residuals and a test for goodness-of-fit.[18] The assessment of the effect of an experimental arm relative to standard and experimental arms relative to one another will be made via the HRs (and associated p-values) derived from this model.

### Secondary analysis

The analysis of OS and time to local–regional failure will be analysed as for PFS.

Toxicity will be assessed using National Cancer Institute Common Terminology Criteria for Adverse Event (CTCAE) v4.03. Late effects will be monitored. The worst grade will summarised and compared between study arms using the Mann-Whitney U test. HE analysis will be undertaken from the perspective of the UK National Health Service, adhering to the National Institute for Health and Care Excellence reference case to assess cost-effectiveness of the experimental arms compared with current standard. QoL will be assessed using EQ-5D and parametric extrapolation undertaken on the PFS and OS curves to generate lifetime outcomes and compare across study arms in terms of quality-adjusted life year gains.

### Interim analysis

Study data will be reviewed at least annually by an independent data monitoring committee (IDMC). Safety stopping in the experimental arms will be monitored continuously based on grade 4/5 acute radiation toxicity.

## Changes to the protocol after the start of the trial

This document is consistent with ADSCaN study protocol (Version 1.2, dated 20 January 2017). There have been no significant changes to the protocol since the start of the study.

## End of the trial

The study will close 12 months after the last patient has recruited or when 305 PFS events have been observed (whichever is later). The chief investigator and/or the TMG have the right at any time to terminate the study for clinical or administrative reasons. The end of the study will be reported to the research ethics committee and regulatory authority (where applicable) within the required time frames.

## QA programme

The trial is subject to an RT QA programme, which includes pre-trial and on-trial components. Dosimetry audit will be performed in centres where a recent relevant external audit has not been completed. Every attempt will be made to streamline with previous national and international audits. The QA programme for the study is coordinated by the UK RTTQA Group. The details of the programme can be found at the RTTQA website, http://www.rttrialsqa.org.uk.

The pre-trial QA includes:
► Facility questionnaire.
► Outlining benchmark case. Target volumes and OAR are delineated on a CT data set provided. The OAR should be outlined according to the ADSCaN OAR Atlas.
► Planning benchmark case. Planned according to the trial protocol on a preoutlined CT (GTV, CTV, PTV and OAR).

Centres that have previously completed pre-trial QA programmes for the individual trial arms, CHART-ED, IDEAL, I-START and Isotoxic IMRT, may be eligible for QA streamlining.

The on-trial QA includes individual case reviews (ICRs) to monitor protocol adherence:
► Prospective ICR for at least the first patient from each centre for each trial arm. Retrospective ICR for selected cases.

## ETHICS AND DISSEMINATION

Adverse events are collected (according to the CTCAE V.4.0 grading system) at each trial visit, and all adverse events on study are reported to the CRUK CTU Glasgow and followed until they resolve or stabilise. Acute and late radiation toxicities continue to be recorded at each follow-up visit.

## Trial monitoring and oversight

Formal on-site data monitoring activities are performed as part of the ADSCaN study. Each site will receive telephone and on-site monitoring visits during the course of the study.

Data are reviewed by an IDMC (comprising two clinicians and one statistician) and Trials Steering Committee. The TMG coordinates and manages the trial's day-to-day activities and is c of health professionals, a patient representative and members of the direct study team, including the principal investigators for each of the experimental arms.

## Dissemination

Data from all centres will be analysed together and published promptly. Individual participants may not publish data concerning their patients that are directly relevant to questions posed by the trial until the TMG has published its report. The TMG will form the basis of the Writing Committee and advise on publications. The trial will be publicised at regional and national conferences. The final results will be presented at scientific meetings and published in a peer-reviewed journal (authorship will be according to the journal's guidelines). In addition, a lay summary of the results will be produced for interested parties, for example, CRUK. The CTU is committed to furthering research by sharing deidentified individual patient data (IPD) from its studies with others in the field who wish to use the data for high-quality science and we are happy to consider proposals from researchers and will share IPD to the maximum extent.

### Author affiliations

[1]Department of Clinical Oncology, Weston Park Hospital, Sheffield, UK
[2]Cancer Research UK Clinical Trials Unit, Institute of Cancer Sciences, University of Glasgow, Glasgow, UK
[3]Division of Cancer Sciences, University of Manchester, Manchester, UK
[4]Guy's & St. Thomas' NHS Trust, Guy's Hospital, London, UK
[5]Velindre Cancer Centre, Cardiff, UK
[6]Department of Physics, Clatterbridge Cancer Centre, Bebington, UK
[7]National Radiotherapy Trials Quality Assurance (RTTQA) Group, Mount Vernon Hospital, Middlesex, UK
[8]Health Economics and Health Technology Assessment, Institute of Health and Wellbeing, University of Glasgow, Glasgow, UK
[9]National Cancer Research Institute Consumer Liaison Group, London, UK

**Acknowledgements** The authors acknowledge the support of the study sponsor NHS Greater Glasgow and Clyde, study funder CRUK and the patients who consent to enter the study. In addition we also acknowledge the individual contributions of Joe Maguire (The Clatterbridge Cancer Centre, Wirral), Emma Parsons, Elizabeth Miles (National Radiotherapy Trials Quality Assurance (RTTQA) Group, Mount Vernon Hospital, Middlesex, UK); Joanne McGarry (NHS Greater Glasgow and Clyde); Stephen Clark, Lindsey Connery, Calum Innes (CRUK Clinical Trials Unit, Institute of Cancer Sciences, University of Glasgow, UK).

**Contributors** MQFH conceived of the study. MQFH, CF-F, DL, JFL, JF, JP, EM, KAB initiated the study design and CAL, RS and AS helped with implementation. MQFH is the grant holder. JP provided statistical expertise in clinical trial design. All the authors contributed to refinement of the study protocol and approved the final manuscript. TH is the patient representative inputting into study design and implementation.

**Funding** This research is funded by Cancer Research UK's (CRUK) Clinical Trials Awards and Advisory Committee (CTAAC). Grant reference number A16604. NHS Greater Glasgow and Clyde (Greater Glasgow Health Board: J B Russell House, Gartnavel Royal Hospital, 1055 Great Western Road, Glasgow, G12 0XH). The sponsor is responsible for confirming there are proper arrangements for the initiation and management of the study. The majority of the sponsor's responsibilities have been delegated to the Chief Investigator (CI) who performs these via the CRUK CTU as the coordinating centre for the trial. As such, the main role of the sponsor is to ensure that the CI and CRUK CTU fulfil their responsibilities as outlined in the Clinical Trial Agreement and to ensure that any identified 'risks' either have controls or action points put in place.

**Competing interests** None declared.

**Patient consent for publication** Not required.

**Ethics approval** West of Scotland Research Ethics Committee, West of Scotland REC 1, 8 November 2016 (REC reference: 16/WS/0165).

**Provenance and peer review** Not commissioned; externally peer reviewed.

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
