## [Reviewer comments · BMJ Open]

ARTICLE DETAILS

TITLE (PROVISIONAL)	Protocol for A Randomised Phase II study of Accelerated, Dose escalated, Sequential Chemo-radiotherapy in Non-Small Cell Lung Cancer (ADSCaN)
AUTHORS	Hatton, Matthew; Lawless, Claire; Faivre-Finn, Corinne; Landau, David; Lester, Jason; Fenwick, John; Simões, Rita; McCartney, Elaine; Boyd, Kathleen; Haswell, Tom; Shaw, Ann; Paul, James

VERSION 1 – REVIEW

REVIEWER	Charles A Butts Cross Cancer Institute University of Alberta Edmonton, Alberta, Canada
REVIEW RETURNED	04-Dec-2017

GENERAL COMMENTS	I would be helpful to know how "not suitable for concurrent chemo-radiotherapy" is defined/determined.
--

REVIEWER	Nagla Abdel Karim The University of Cincinnati, USA
REVIEW RETURNED	17-Jan-2018

GENERAL COMMENTS	1. In the era of immunotherapy, would the use of consolidation immunotherapy (Given the data from Durvalumab) be of consideration in the future. Can this be included at least in the future discussion and consideration?2. The choice of systemic chemotherapy should be more clarified. The choice of platinum doublet will have variable outcomes in non-squamous versus squamous cell lung cancer given data of pemetrexed and other therapies used in combination with platinum.3. The study is an excellent option for patients who are not eligible for concomitant chemoradiation. There should be specific criteria for this group of patients. In the inclusion criteria patients with ECOG PS 0 and adequate PFTs will be included but it was unclear of what will be the criteria of ineligibility to concomitant chemoradiation if it were due to cardiac conditions or other contraindications.
--

REVIEWER	Hong Ryull Pyo Samsung Medical Center, Republic of Korea
REVIEW RETURNED	21-Mar-2018

GENERAL COMMENTS	This manuscript is not a result of a protocol, but a protocol itself for a phase II study of sequential chemo-radiotherapy (SCRT) with accelerated and dose-escalated radiotherapy (RT) schedules in
--

	non-small cell lung cancer. This protocol seems to be already opened since 22 August 2017 in UK. The aim of this study is to pick the winner among various RT schedules which have been completed in UK by comparing them with a standard RT schedule. This protocol is organized and written decently and therefore I have little to criticize on the protocol structure and format, but there are several issues on study design to be discussed.  1. The authors argue that most patients are not suitable for the concurrent chemo-radiotherapy (CCRT), and there seems to be one reference for this notion according to the current manuscript (De Ruyscher et al, Ann Oncol 20:98-102, 2009). However, this notion ("most are not suitable for CCRT") cannot be generalized yet and even the authors of the above reference stated in the discussion that their suggested criteria for unsuitability for CCRT might be too wide. 2. The authors also stated that development of RT techniques such as IMRT can provide the potential to escalate the RT dose to the target while decreasing the dose to normal tissues for their SCRT schedules. However, this is also true for CCRT and nowadays many more patients can be safely treated with CCRT with significantly less toxicity than before with the help of IMRT, image guided radiotherapy and respiratory guided radiotherapy, etc. 3. To summarize, superiority of CCRT over SCRT is historically confirmed by several randomized trials and therefore majority of the patients should still be treated with CCRT to get the superior result, with the help of modern RT technologies to lessen the toxicity. 4. In conclusion, it is recommended that the eligibility criteria for this protocol should be confined to the patients who are really unsuitable for CCRT. The current criteria for unsuitability for CCRT are not specified in the protocol and many patients who are eligible for this protocol seems to be also eligible for CCRT.
--	--

REVIEWER	Ramses Sadek Augusta University, Georgia Cancer Center, USA
REVIEW RETURNED	20-Apr-2018

GENERAL COMMENTS	I have few issues with the statistical methods and power calculation:  1. There are 11 factors for stratification without defining categories within each factor. These are too many factors to be considered, specially if you have only 60 patients in each of the experiment arms. 2. Using a significant level at 1-sided test of 20% is too high. It should never be more that 10% (which is equivalent to 0.20 in two-sided test). That is a too high risk approach. 3. The effect size (33% increase in median PFS) is not based on data or literature review but made up without any ground as the authors stated that it is the UK requirement. 4. There is no detailed analysis methods or handling of multiplicity, missing data, or specification of datasets/population used in each analysis.
--

VERSION 1 – AUTHOR RESPONSE

Reviewer(s) Comments to Author:

Reviewer: 1

Reviewer Name: Charles A Butts

Institution and Country: Cross Cancer Institute, University of Alberta, Edmonton, Alberta, Canada

Please state any competing interests: None declared

Please leave your comments for the authors below

It would be helpful to know how "not suitable for concurrent chemo-radiotherapy" is defined / determined.

We fully agree with the reviewer that it would be helpful to clarify the definition of suitability for concurrent chemo-radiotherapy. The challenge of defining those suitable for concurrent rather sequential treatment has exercised the TMG throughout the design and development of the protocol. During that process we referred to national / international guidelines an example of which is the 2010 British Thoracic Society guidelines –

'The literature supports the use of concurrent chemoradiotherapy in patients aged <75 years (patients aged >75 years have not been included in clinical trials), performance status 0-1, with reasonable lung function, without major comorbidities, and for whom the radiotherapy plan produces acceptable normal tissue doses'

More recent guidelines have given even less definition -

NCCN 2015 'Frail patients may not tolerate concurrent CRT'

ESMO 2017 'Concurrent CRT is considered the preferred treatment for patients who are fit'

This means that with these poorly defined selection criteria practice is greatly influenced by local experience, hence the inclusion criteria in the ADSCAN protocol 'Inoperable disease, unsuitable for concurrent chemoradiation, in the opinion of the treating Oncologist' The recent publication by Miller et.al. indicates that there is a population of patients for whom the concurrent vs sequential approach may be detrimental.

Miller ED, Fisher JL, Hagland KE, Grecula JC, Xu-Welliver M, Bertino EM et al. The addition of chemotherapy to radiation therapy improves survival in elderly patients with stage III NSCLC. JTO 2018;13:426-435

Reviewer: 2

Reviewer Name: Nagla Abdel Karim

Institution and Country: The University of Cincinnati, USA Please state any competing interests:

Speaker for Merck, Genetech

Please leave your comments for the authors below

1. In the era of immunotherapy, would the use of consolidation immunotherapy (Given the data from Durvalumab) be of consideration in the future. Can this be included at least in the future discussion and consideration?

We are aware that the standard of care for the systemic treatment of stage III NSCLC may evolve over the course of this study and in particular the data coming from the Pacific study is exciting and is likely to prove practice changing over the coming years. However, the current evidence derived is for maintenance immunotherapy following concurrent chemoradiotherapy treatment rather than sequential so the TMG / DMEC will be closely monitoring the presented / published data and will amend the recommendations for systemic treatment in the protocol in the light of the evolving evidence.

We have amended the standard of care paragraph to reflect this.

2. The choice of systemic chemotherapy should be more clarified. The choice of platinum doublet will have variable outcomes in non-squamous versus squamous cell lung cancer given data of pemetrexed and other therapies used in combination with platinum.

We feel the current standard for chemotherapy for NSCLC is the use of a platinum based doublet regime and have stated this as one of the eligibility criteria - *Patients who have had a partial response or stable disease following 3 cycles of platinum based chemotherapy.*

Clinicians can select the 2nd non-platinum agent on a histological basis and according to standard practice at their site. Within the protocol the recommended non-platinum agents are pemetrexed or vinorelbine for non-squamous carcinoma and gemcitabine or vinorelbine for squamous carcinoma. These are the agents used for > 90% of current UK practice.

We have amended the standard of care paragraph to clarify the recommended chemotherapy regimens.

3. The study is an excellent option for patients who are not eligible for concomitant chemo-radiation. There should be specific criteria for this group of patients. In the inclusion criteria patients with ECOG PS 0 and adequate PFTs will be included but it was unclear of what will be the criteria of ineligibility to concomitant chemo-radiation if it were due to cardiac conditions or other contraindications.

Thank you, as discussed in response to reviewer 1 the fitness criteria for concurrent treatment is poorly defined which limits the ability to lay down specific criteria in the protocol. In practice the decision is usually made after detailed risk / benefit discussion of the concurrent vs sequential approach with the patient.

Reviewer: 3

Reviewer Name: Hong Ryull Pyo

Institution and Country: Samsung Medical Center, Republic of Korea Please state any competing interests: None declared

Please leave your comments for the authors below

This manuscript is not a result of a protocol, but a protocol itself for a phase II study of sequential chemo-radiotherapy (SCRT) with accelerated and dose-escalated radiotherapy (RT) schedules in non-small cell lung cancer. This protocol seems to be already opened since 22 August 2017 in UK.

The aim of this study is to pick the winner among various RT schedules which have been completed in UK by comparing them with a standard RT schedule.

This protocol is organized and written decently and therefore I have little to criticize on the protocol structure and format, but there are several issues on study design to be discussed.

Thank you, this is indeed a protocol publication and the ADSCaN study is now open and recruiting. Therefore, we are limited in the changes we can make to protocol and are grateful for the comments received from the reviewers which we use to inform some planned amendments that we will be submitting for research ethics committee approval over the coming weeks.

1. The authors argue that most patients are not suitable for the concurrent chemo-radiotherapy (CCRT), and there seems to be one reference for this notion according to the current manuscript (De Ruysscher et al, Ann Oncol 20:98-102, 2009). However, this notion ("most are not suitable for CCRT") cannot be generalized yet and even the authors of the above reference stated in the discussion that their suggested criteria for unsuitability for CCRT might be too wide.

We have re-written this paragraph and provided additional references.

2. The authors also stated that development of RT techniques such as IMRT can provide the potential to escalate the RT dose to the target while decreasing the dose to normal tissues for their SCRT schedules. However, this is also true for CCRT and nowadays many more patients can be safely

treated with CCRT with significantly less toxicity than before with the help of IMRT, image guided radiotherapy and respiratory guided radiotherapy, etc.

3. To summarize, superiority of CCRT over SCRT is historically confirmed by several randomized trials and therefore majority of the patients should still be treated with CCRT to get the superior result, with the help of modern RT technologies to lessen the toxicity.

We agree that the technological advances can and should be used to offer concurrent chemo-radiotherapy more safely to a wider group of patients. However, we would argue that this study primary aim is to explore the relative advantages of different dose-escalation schedules for UK patients who ordinarily would receive sequential CRT in the populations where the increased toxicity of the concurrent approach is considered significant.

4. In conclusion, it is recommended that the eligibility criteria for this protocol should be confined to the patients who are really unsuitable for CCRT. The current criteria for unsuitability for CCRT are not specified in the protocol and many patients who are eligible for this protocol seems to be also eligible for CCRT.

See responses to reviewer 1 and 2, we acknowledge 'local experience' of CCRT may lead to some differences in patient characteristics between different centres and the stratification factors will be used by the CTU to ensure balance across the arms.

Reviewer: 4

Reviewer Name: Ramses Sadek

Institution and Country: Augusta University, Georgia Cancer Center, USA Please state any competing interests: None declared

Please leave your comments for the authors below

I have few issues with the statistical methods and power calculation:

1. There are 11 factors for stratification without defining categories within each factor. These are too many factors to be considered, especially if you have only 60 patients in each of the experiment arms.

Controlling imbalance is especially important in a small study and minimisation is being used as the technique to stratify patients. The number of stratification factors has no impact on the effectiveness of this approach and we have run many studies with a large number of stratification factors using this technique. We have simulated the study to test the approach and found that it works well and produces good balance.

2. Using a significant level at 1-sided test of 20% is too high. It should never be more than 10% (which is equivalent to 0.20 in two-sided test). That is a too high risk approach.

The selection of 20% 1-sided is based upon:-

Design Issues of Randomized Phase II Trials and a Proposal for Phase II Screening Trials Lawrence V. Rubinstein, Edward L. Korn, Boris Freidlin, Sally Hunsberger, S. Percy Ivy, and Malcolm A. Smith Journal of Clinical Oncology 2005 23:28, 7199-7206

These authors are all from Cancer Therapy Evaluation Program / National Cancer Institute and their publications deem 20% acceptable and we would fully agree that you should go beyond that. We have added this to the references.

3. The effect size (33% increase in median PFS) is not based on data or literature review but made up without any ground as the authors stated that it is the UK requirement.

Choosing an effects size is to a certain extent always a matter of judgement. There are no data available that would allow us to reliably estimate the likely impact of these new radiotherapy schedules (there is only uncontrolled small scale safety study data available for each). The investigators made a

judgement as to what effect size may change clinical practice and is realistic. This effect size chosen corresponds to hazard ratio of 0.75, a fairly typical target in phase III trials. As noted in the protocol the recent US study (RTOG 0617) targeted a larger improvement of a 40% increase.

4. There are no detailed analysis methods or handling of multiplicity, missing data, or specification of datasets / population used in each analysis.

We would not routinely put all this detail in a study protocol. However, as per our Clinical Trials Unit Standard Operating Procedures, we will produce a detailed Statistical Analysis Plan prior to any data analysis.

VERSION 2 – REVIEW

REVIEWER	Hong Ryull Pyo Samsung Medical Center, Seoul, Korea
REVIEW RETURNED	08-Aug-2018

GENERAL COMMENTS	No more questions for the manuscript
--------------------------------------

REVIEWER	Ramses Sadek, PhD Augusta University, Georgia Cancer center, USA
REVIEW RETURNED	31-Jul-2018

GENERAL COMMENTS	Adequate biostatistics sectio
-------------------------------